# Soursop (*Annona muricata*) Properties and Perspectives for Integral Valorization

**DOI:** 10.3390/foods12071448

**Published:** 2023-03-29

**Authors:** Ivone Lima Santos, Antonio Manoel da Cruz Rodrigues, Edna Regina Amante, Luiza Helena Meller da Silva

**Affiliations:** Programa de Pós-Graduação em Ciência e Tecnologia de Alimentos (PPGCTA) [Graduate Program in Science and Food Technology], Universidade Federal do Pará (UFPA), Belém 66075-110, Pará, Brazil

**Keywords:** *Annona muricata*, bioinsecticides, biorefinery, pharmaceuticals, circular bioeconomy, innovation, phytochemicals

## Abstract

The increased international interest in the properties of soursop (*Annona muricata*) alerts us to the sustainability of productive chain by-products, which are rich in phytochemicals and other properties justifying their industrial application in addition to reducing the environmental impact and generating income. Chemical characteristics of soursop by-products are widely known in the scientific community; this fruit has several therapeutic effects, especially its leaves, enabling it to be used by the pharmaceutical industry. Damaged and non-standard fruits (due to falling and crushing) (30–50%), seeds (3–8.5%), peels (7–20%), and leaves, although they constitute discarded waste, can be considered as by-products. There are other less cited parts of the plant that also have phytochemical components, such as the columella and the epidermis of the stem and root. Tropical countries are examples of producers where soursop is marketed as fresh fruit or frozen pulp, and the valorization of all parts of the fruit could represent important environmental and economic perspectives. Based on the chemical composition of the fruit as well as its by-products and leaves, this work discusses proposals for the valorization of these materials. Soursop powder, bioactive compounds, oil, biochar, biodiesel, bio-oil, and other products based on published studies are presented in this work, offering new ideas for opportunities for the regions and consumers that produce soursop.

## 1. Introduction

Soursop (*Annona muricata*) is a tropical fruit belonging to the *Annonaceae* family, native to tropical countries, also known according to the producing countries as *graviola*, *guanabana*, *sauersak*, *guayabano*, and several other regional names [1]. It has economic importance and growth in the Caribbean as well as the equatorial belt of the Americas, mostly in the Bahamas, Bermuda, Cuba, Dominican Republic, Grenada, Mexico, Costa Rica, St. Vincent, Puerto Rico, Colombia, Venezuela, Equator, and Brazil. It is distributed throughout the tropics of the world, including the Caribbean, Africa, and Southeast Asian countries such as Thailand, Malaysia, Indonesia, and the Philippines, with Mexico as the principal producer country [2,3].

According to Lima and Alves [4], it is difficult to obtain accurate data regarding production statistics, but they indicate that Mexico, Puerto Rico, Venezuela, and Costa Rica are the largest exporters of soursop products, mainly as frozen pulp and fresh fruit. Brazil, Colombia, and Venezuela are the main producers of soursop in South America, with Venezuela being the largest producer in this region. Mexico leads world production, while Brazil is in third position. In these countries, the consumption of fresh fruit and frozen pulp is predominant [4].

Considering the properties of the *Annonaceae* family, along with its cultivation and origin in tropical regions, there is a growing interest in cultivating it in Europe as well, which is attributable to climate changes resulting from global warming, making some regions suitable for the cultivation of the fruit. In Europe, Spain is the most important producer of cherimoya fruit. In Italy, *A. cherimola* is well adapted to the pedoclimatic conditions of the Tyrrhenian coastal areas of Sicily and Calabria [5].

The fresh fruit of soursop is a favorite on the market due to its pleasant and sweet taste, but the sensitive characteristics of the fruit lead to the production of the pulp as the most economically attractive commercial form. Sacramento et al. [6] report pulp yield of 83.12% to 85.85% and Nolasco-González, Hernández-Fuentes, and González [7] report values between 46.8% and 80.6%. This yield difference described by the authors is attributed to changes according to the place where the fruit is produced. The pulp is firm and soft, and this enables its second form of commercialization as frozen pulp; from this we can get the secondary products, such as juice, nectar, ice cream, jam, and yogurt [8,9]. The main products found in the world market are frozen pulp, soursop powder, and dried soursop leaves. The concentration of production in these tropical countries justifies an article gathering information on the potential of all parts of the fruit, including the leaves, as residues from management in orchards.

Producer countries find a common problem in the management of fresh fruit, with moisture in the range of 80% [10] and sensitivity to changes during transport. In addition, challenges in the cultivation of soursop, such as fruit disease (fruit borer), represent a loss corresponding to 30% of the pulp, which can result in a total pulp yield of 54% [11], meaning a great loss of raw material in this production chain.

The entire soursop production and commercial process generates a significant amount of waste during processing for pulp production, crushed fruits that fall into the orchard, and ripe or unripe fruits, plus the peels and the seeds. During planting with tree pruning, many leaves are removed [12]; in harvesting, transport, and fresh commercialization with excessive ripening, fruits can fall and be crushed, causing the loss of the entire fruit [4]. Baddrie and Schauss [1] highlight that post-harvest losses of fresh fruit can reach up to 75.8% due to a lack of knowledge about losses and how to prevent them in some producing regions. Figure 1 illustrates the most important waste generation points in the soursop production chain.

Considering the waste of the soursop production chain, finding and suggesting applications for this waste reduces undue disposal in the environment in the form of solid waste, and it also represents a financial opportunity for the producing regions [13,14]. Based on published works on the properties of soursop residues (standard-out fruits, peels, seeds, leaves, and others), the main objective of this work is to offer information that can contribute to future biorefinery projects and to the circular economy in the soursop production chain.

## 2. Characteristics of Soursop By-Products

The genus *Annona* has 70 species, with *Annona muricata* being the best known. *A. bonplandiana* Kunth, *A. cearensis* Barb. Rodr., *A. macrocarpa* Werckle, *A. muricata* var. *borinquensis* Morales, and *Guanabanus muricatus* M. Gómez are other known species [15]. Detailed reviews on the botanical characteristics of *Annona muricata* can be found in the works of Coria-Téllez et al. [3] and Behl et al. [16].

Just as the different varieties interfere with the properties of the pulp, it is important to consider these differences in terms of the residues in the soursop production chain. The main soursop varieties grown in Brazil are Crioula, Blanca, FAO II, Lisa, and Morada, with significant differences regarding the fruit size, shape, yield, and pulp properties [10,15]. One of the main differences between the fruits is the soursop fresh fruit weight, which can fluctuate from 2.39 kg to 10 kg according to the variety and the growing conditions [6,11]. Pinto et al. [15] state the average weight is 4.0 kg, but in Mexico, Venezuela, and Nicaragua it ranges from 0.4 kg to 1.0 kg [3].

Soursop pulp, leaves, peels, and seeds are rich in phenolic compounds that give the fruit numerous health benefits [17,18] (Table 1), which makes them potential sources for the extraction of bioactive compounds that can be used in the pharmaceutical, cosmetic, and food industrial sectors [19]. These compounds attribute functionality and value to products developed with soursop by-products.

Important phytochemicals are found in soursop pulp, with differences in the phenolic compounds founded according to the kind of extraction, whether conventional [17] or ultrasound assisted [19]. The biological properties evaluated in in vitro and in vivo studies that stand out are the antidiarrheal [26], antidiabetic [27,28], antifungal [29], antihypertensive [28], anticancer, and antimicrobial [30] ones, plus the antioxidant properties [30,31].

Due to their pharmaceutical applications, the leaves have a wide spectrum of phytochemical compounds identified in different studies. Some of those are limited to phenolic compounds in plants cultivated in a certain region [22], with others signaling only antiangiogenic compounds by metabolomics [21] or as source of antioxidant compounds [23]. The results of in vivo and in vitro studies on biological properties found in the leaves compared to other residues are highlighted in numerous works, justifying the consolidated national and international market for soursop leaves.

Since it represents a smaller percentage among soursop by-products, the peel has been less studied, but it also contains important phytochemical compounds with an emphasis on resorcinol [24], which is not found in works on the other parts of the plant. As for the biological properties, this same work highlights the restorative effect of pancreatic cells. Other works emphasize the biological activities already found in the pulp and leaves.

Phytochemical compounds in the seeds are many of those present in the pulp, leaves, and peel. However, considering that the seeds have the oil fraction as the predominant one, other lipophilic compounds can be found that expand the properties of this byproduct compared to other parts of the plant, such as, for example, attenuation in benign prostatic hyperplasia [25].

Table 2 provides a general overview of the bioactive properties already identified in in vivo and in vitro studies on soursop pulp, leaves, seeds, and peel.

## 3. Food Products from Soursop By-Products

The applications of by-products for the preparation of new products depend on their characteristics and proper care, incorporating these materials into the circular economy of the soursop production chain.

Crushed, defective fruits that fall into the orchard have no commercial value; they are not even introduced into the industrial process and this reality also occurs in the soursop production chain. The main challenge when marketing fresh fruit is that fresh soursop has an accelerated maturation process, with a shelf-life of approximately five days post-harvest [10,11], reaching up to 60% post-harvest loss in Mexico due to the perishable fragility of the fruit [44]. In addition to other factors associated with sensitivity, crushing can occur due to the fall of overripe and unripe fruit from trees or during transport and exposure on the shelves, turning the crushed fruit into residues of the production chain [4].

Regardless of these features, even though they cannot be included in the processing, mashed fruits still can be used, increasing the consumption of soursop and reducing these by-products, as they still have functional properties (total phenolic of 2886,60 mg GAE/100 g dry basis) [45].

Despite representing a small percentage (3 to 8,5%) [46] of the waste generated in this production chain, seed soursop composition can be compared to important commercial sources of lipids and proteins. Dehydrated seeds may have 8.5% moisture, 2.4% protein, 13.6% ash, 8.0% fiber, 20.5% fat, and 47.0% carbohydrate [47]. The assessment performed by Aguilar-Hernández et al. [19] indicated that soursop seeds and integument contain 27.34% and 11.4% of carbohydrate, 4.36% and 24.69% of protein, 2.29% and 2.58% of ash, and 43.44% and 63.32% of fiber, respectively. The carbohydrate and fiber contents were higher than the ones observed in *Annona squamosa*, while 22.57% of lipids were found only in the seed, in addition to iron, zinc, copper, potassium, sodium, calcium, and magnesium. Considering this important composition, the seeds can be used in different processes and products.

The soursop fruit peel has not been as studied as the leaf, but this part, considered as waste (20%) [46], also has a profile of bioactive compounds (Table 1) allowing its valorization and its technological use.

Despite the limited work on the use of soursop peels in new products, they can be used in processes similar to other soursop residues, such as in the use of infusion currently implemented by traditional medicine and their resultant pharmaceutical application [48] in extracts [19,49], animal feed production, and compost.

Soursop leaves are generated as residues from the management of orchards in the pruning process, with a great contrast between the generation of this residues and the research works presenting the pharmacological properties of the leaves, being widely used in producing regions and found in the dehydrated form in the international market, both commercialized as dry leaves and in encapsulated powder.

Each of these by-products of the soursop production chain can be used for different applications as new products, according to their properties.

### 3.1. Fermented Products

The production of a fermented alcoholic beverage (called wine by some authors) from pure or blended fruit is a common practice. The production of a tropical fruit fermented alcoholic beverage represents a promising alternative for the valorization of these raw materials [50,51]. Okigbo and Obire [52] (2008) and Ho et al. [53] have proved the potential of soursop for fermented alcoholic beverage production, obtaining products with acceptable sensory characteristics of color, flavor, and aroma, indicating the use of these discarded fruits from the soursop production chain.

Kombucha drink is gaining ground among consumers. It is traditionally produced from the fermentation of tea and sugar through symbiotic cultures [54]. In an attempt to innovate and incorporate flavors, the inclusion of fruit juice, such as soursop, was studied in the preparation of kombucha.

Tan et al. [55] developed kombucha using pasteurized soursop juice, black tea, and sugar by fermentation (SCOBY—symbiotic community of bacteria and yeast). Despite the work previously mentioned, more studies with soursop pulp in the preparation of kombucha are required, thus expanding the evaluations and comparing it with kombucha from traditionally consumed fruits. Soursop leaves can be further studied in future works, replacing tea in the production of kombucha.

From acetic fermentation, Isham et al. [56] (2019) and Ho et al. [57] developed vinegar from soursop, resulting in a vinegar with a less acidic taste and a sweeter characteristic compared to commercial (apple) vinegar, which has 4.4 pH, 3% sugar, 1.7% reducing sugar, 7.0% ethanol, and 3.5% acetic acid. Thus, in addition to the use of soursop in the production of vinegar in order to reduce waste, it is also possible to use techniques favoring the improvement of this production and generation of other fermentation products. 

### 3.2. Dried Products

Just as the pulp of selected fruits is dehydrated and marketed as soursop powder, the pulp of non-standard fruits can be transformed into powder. Therefore, dehydrated discarded leaves and fruits could represent a new opportunity for the recovery of this residues.

Crushed soursop fruits were dehydrated by spray drying and the resulting powder product did not show significant differences in its composition compared to fresh pulp, preserving total carotenoids (6.79 µg g^−1^), total phenolic (158.95 mg of GAE/100 g), and flavonoids (85.17 mg of quercetin 100 g^−1^) as well as aroma components, and it also showed antioxidant activity [58].

The lyophilized pulp with 18% maltodextrin resulted in soursop powder with greater brightness compared with the control [59]. Telis-Romero et al. [60] also performed the pulp drying in a fluidized bed with the same amount of maltodextrin. Thus, it is possible to obtain soursop powder using various drying methods, but it is important to choose the one that can best maintain the nutritional, aromatic, and microstructural qualities of soursop as well as the one that best fits into the reality of the producer with the responsibility of offering the best product from re-use, enabling the powder to be used in other sectors of the industry. Dehydrated fruit is offered in a wide range of applications, transforming fruits that would go to compost, when well sorted and with all the required sanitary care, into powdered soursop.

A common application for fruit peel is the production of flour and/or powder, allowing its use in various products and compound extractions. Although the flour is prepared with other parts of the fruit [61], this process can be adapted to the peels and used in bakery products, even at low concentrations, allowing the reduction of this residues and the enrichment of food products. The use of fruit residues flour, such as soursop peel, becomes an important procedure for the incorporation of these compounds, and their increased dietary fiber, common in fruit peel, contributes to good health [62].

Soursop peel flour can allow phenolic compounds to be incorporated into products, increasing their antioxidant capacity and acting as an important nutraceutical [49].

### 3.3. Tea

Soursop leaves are traditionally consumed by infusion in folk medicine as an anticancer, analgesic, and antispasmodic agent. It is internationally marketed for therapeutic purposes and can be used with herbal mixtures [63,64].

The aqueous leaf extract contains the following phytochemicals: alkaloids, saponins, tannins, phenols, phytosterols, terpenoids, and anthraquinones in high amounts, as well as cardiac glycosides, coumarins, lactones, and flavonoids, thereby justifying its traditional use as a medicine [65].

The soursop leaves decoction contains caffeic acid (30.0 µg/100 mL of infusion) and isomers of chlorogenic acids (3-caffeoylquinic acid-196 µg/100 mL, 4-caffeoylquinic acid-124 µg/100 mL, 5-caffeoylquinic acid-9 µg/100 mL, 3,4-dicaffeoylquinic acid-14.2 µg/100 mL, 3,5-dicaffeoylquinic acid-24.2 µg/100 mL, and 4.5-dicaffeoylquinic acid-6.4 µg/100 mL) [18].

The aqueous extracts of the soursop leaves have antioxidant capacity due to their phytochemical constituents and pharmacological properties, which are also commonly used in infusions as a soothing, anti-inflammatory, antiallergic, antibacterial, and antiviral agent, as well as for fever, pain, and diarrhea. The alkaloids are one of those constituents responsible for such characteristics. The presence of alkaloids widely used in medical science justifies the pharmacological effects experienced by people after tea consumption [3,65].

Despite the studies on the toxicological effects of soursop and its leaves influencing atypical Parkinson’s disease [66], for example, the amounts used for decoction are not sufficient to harm health. There are also differences between drinking tea leaves and the use of a given isolated compound, as this can increase the bioavailability. Furthermore, the studies seem somewhat contradictory, which generates some controversy regarding the definition of the harmful effect to the human organism [48]. However, with respect to questions about the therapeutic versus the toxicological potential, the excessive consumption is not indicated.

In view of the above and beyond the examples elucidated, it is possible to observe great potential and influence of the leaves for application in the pharmaceutical field, which is explored and emphasized in the literature [67,68]. There are studies reporting the use of soursop leaves to get tea for the development of kombucha, for example [69,70].

### 3.4. Essential Oil and Oleoresin

Fruits with lower added value were studied as sources of essential oils. Gyesi et al. [71] obtained essential oil from fresh soursop pulp with a yield of 0.11%. Dozens of compounds were identified, including: 2-hexenoic acid methyl ester, β-caryophyllene, 1,8-cineole, linalool, Ç-sitosterol, 2-hydroxy-1-(hydroxymethyl) ethyl ester, 2-propenoic acid, 3-phenyl-, and methyl ester as the most abundant ones [71,72].

Essential oils can also be extracted from soursop leaves, with a yield of 0.67% and showing light-yellow color [71], thus increasing the range of opportunities in the development of products with soursop culture. As with the other products, the essential oil of the soursop leaf has an antioxidant capacity as well as 80 different compounds [73].

In addition to the essential oil, it is possible to obtain oleoresin; however, it differs in physical and chemical terms in liquid form under environmental conditions, while oleoresin may contain waxes in solid or semi-solid form [74]. Soursop leaf oleoresin has bactericidal activity containing 22.7% of phytosterols as the main class of compounds [75].

### 3.5. Oil

From soursop seeds, it is possible to obtain 22.57% to 34.61% of oil [67], the fatty acid profile of which consists of 10,13-octadecadienoic, myristic, palmitic, palmitoleic, stearic, oleic, linoleic, α-linolenic, elaidic, methyl-palmitic, and arachidic acids, of which 27.6% are saturated and 70.0% are unsaturated fatty acids [20,76].

Soursop seed oil has 2.21 mg/kg of α-tocopherol, 7.1 mg/kg of γ-tocopherol, 18.0 mg/100 g of campesterol, 47.2 mg/100 g of stigmasterol, 85.5 mg/100 g of β-sitosterol, 0.8 mg/100 g of *ρ*-coumaric acid, 86.2 mg/100 g of epicatechin, 4.4 µg/g of β-carotene, and 13.98 mg EAG/100 g of phenolic totals as bioactive compounds, an antioxidant capacity of 22.7 mg/mL, 79.1 μmol Trolox/100 g and 236.4 μmol FeSO_4_/100 g for DPPH, ABTS, and FRAP, respectively, exhibiting 27.5 h of oxidative stability [77].

As the oil content in the seed is higher than 20% [78], the extraction can be achieved by cold pressing, which better preserves the properties. The first oil is obtained by cold pressing dry seeds or by using microwave pre-treatment. The cake can proceed to a second extraction process, resulting in greater oil recovery. The CO_2_ supercritical extraction from the cake is promising not only for the yield, but also for the maintenance of bioactive compounds in the raw material [79,80].

Other methods to obtain the oil were also studied, such as extraction by Soxhlet with ethyl ether, with a yield of 30.72%; Bligh-Dyer, using methanol and chloroform after seed preparation, with drying and grinding [14,81]; and ultrasound assisted extraction and enzymatic aqueous extraction (although the latter has a low yield (11.15 g/100 g of seeds) [82].

Seed oil has beneficial effects on human health, such as on diabetes (type 1), producing an antihyperglycemic effect [81,83].

### 3.6. Starch

Expanding the application possibilities to crushed soursop fruits, there are some perspectives for the production of starch from these residues. Nwokocha and Williams [84] used soursop pulp for starch extraction, achieving 27.30% of starch yield. This is indicated here for the development of future work about the application of this new source of starch.

## 4. Non-Food Applications for Soursop By-Products

Under the concepts of circular economy [85], not all by-products of the food industry are applicable to the sector itself. However, a look at the full valorization of raw materials could contribute to increasing income and reducing the environmental impact on the food sector, in addition to generating alternatives to non-renewable sources for new products. Therefore, the by-products of the soursop production chain can be valued in other sectors, according to the examples presented in this document.

### 4.1. Pharmaceutical Products

The production of metallic nanoparticles with soursop leaf extract has several therapeutic applications that can be explored, such as antidiabetic, antimicrobial, and antioxidant usage, as well as the inhibition of lipid peroxidation activity [36]. The use of plant extracts in the production of nanoparticles is an interesting proposition, since constituent compounds such as polyphenols, tannins, flavonoids, and terpenoids act as reducers, coating agents, and stabilizers in the synthesis of nanoparticles, being named as green-synthesized nanoparticles [86]. Therefore, the use of plant extracts such as *Annona muricata* in the synthesis of nanoparticles such as silver is considered an environmentally-friendly and low-cost process [87].

With respect to the health area, specifically the anticancer effect, Meenakshisundaram et al. [88] used the aqueous extract of leaves for the synthesis of silver nanoparticles as an anticancer agent. These were evaluated at the molecular level, where the use of the crude leaf extract was necessary to have a concentration of 120 µg/mL for 50% inhibition (IC50) of cell viability. In the nano form, the anticancer activity was increased, whereas only 6 µg/mL was needed to inhibit 50% of the lung cancer cell (A549) growth. This is a dose-dependent effect, as mentioned by Gavamukulya et al. [89]. Silver nanoparticles with leaf extract have high antioxidant capacity for DPPH and ABTS, with 50% inhibitory concentration (IC50) of 51.80 µg/mL and 30.78 µg/mL, respectively. They also act in the α-amylase enzymes and α-glycosidase inhibition involved in carbohydrate metabolism [36]. Despite the approach to leaves in the preparation of nanoparticles, the fruit extract has also been studied for the synthesis of gold nanoparticles, showing an anticancer effect in vitro [90].

The synthesis and use of metallic nanoparticles with soursop leaf extracts have a wide range of utility for industrial application. In addition, the production of nanoparticles with plant extracts is considered a green process, always alerting us to the characteristics, limits, and environmental effects of the nanoparticles themselves.

The soursop columella, representing 4% of the residual material, has not been studied yet; however, this part can also serve as a source to obtain compounds, as it contains gallic acid, coumaric acid, cinnamic acid, caffeic acid, chlorogenic acid, 4-hydroxybenzoic acid, and neochlorogenic acid [19]. Similar to the root and the stem bark, soursop columella has similar phytochemical screening with tannins, flavonoids, saponins, terpenoids, carbohydrates, reducing sugars, monosaccharides, pentoses, ketoses, starch, protein, arginine, cysteine, aromatic amino acids, phenolic amino acids, alkaloids, steroids, and phenolics, with a difference for the cardiac glycosides compound, which is found in the root bark only [89].

### 4.2. Cosmetics

Unlike the soursop pulp, which is more predominantly used in the food sector, properties of soursop seed oil indicate its use also for cosmetics. As an emollient in the production of cosmetic creams to replace synthetic products, soursop seed oil containing 42.94%–43.73% oleic acid and 29.5–30% linoleic acid favors chemical and physical characteristics of the products [14].

### 4.3. Animal Feed

The use of by-products originating from the food industry has naturally increased and is taking part in the supplementation of animal feed. To that end, soursop seed also has potential for this branch of the industry, since it has antinutritional compounds such as phytate, tannin, and cyanide, but at low concentrations [91,92].

Pinto et al. [93] suggest that the seeds and the cake resulting from oil extraction can be used as a complement in animal feed after drying and grinding, which allows a longer storage time. Biomass contains dry matter (852.5 g/kg of natural material), minerals (17.3 g/kg of dry matter), ether extract (194.6 g/kg of dry matter), crude protein (164.1 g/kg of dry matter), and other important compounds for the animal diet. However, the authors warn about the high concentrations of lipids in soursop cake for ruminant feeding, highlighting the usage of adequate dietary amounts associated with other ingredients. 

### 4.4. Bio-Diesel, Bio-Oil, Bio-Char and Gas

Soursop seed oil has free fatty acid (3.5%) levels that must go through the conversion process to be able to follow the transesterification. Of the fatty acids in the seed, 44.5% are monounsaturated fatty acids and 30.5% are polyunsaturated fatty acids, which contribute to the optimization of the transesterification process, being ideal for converting oil into biodiesel (fatty acid alkyl esters) [94,95] with a yield of 83% to 91% [96].

Another important option for using soursop seeds is the production of bio-oil. The bio-oil obtained from the liquid fraction of the slow pyrolysis of the soursop seed cake has the same biological constituents present in cake, consisting of lignin, cellulose, hemicellulose, fatty acids, and proteins. The bio-oil profile has three furan derivatives, thirteen phenolic derivatives, six phenols, and six aromatic hydrocarbons [96].

The cake or biomass from *Annona muricata* seed oil extraction, as processing residues, also has promising industrial applications. The soursop seed applications are extended to the biochar, a biofertilizer alternative to synthetic ones that has been gaining ground due to its cost-effectiveness and to the corrective properties of the soil influencing the microbial community. Biochar is considered ecologically viable due to the use of residual biomass [97]. It can be produced by industries, small producers, and households in a pyrolysis process through the thermochemical conversion (limitation of oxygen with high temperature) of biomass to sustainable environmental practices [98]. The seed cake subjected to slow pyrolysis (400 °C) results in the production of biochar with a 32.2% yield [96].

Soursop seeds also have potential for gas production. The product gas is one of the artifacts generated through pyrolysis from residual biomass. This gas with hydrogen is one of the goals of pyrolysis and it can be used in spark ignition or compression engines [99,100]. As for the pyrolysis gas from the soursop seed biomass, there are still few in-depth studies on this product. Schoroeder et al. [96] demonstrated the process and products resulting from slow pyrolysis (400 °C) of soursop seed cake, pointing out that the pyrolytic gas has a yield of 17.7%. However, this study emphasized the liquid fractions from pyrolysis.

### 4.5. Biopesticides

The use of insecticides in agro-industrial crops is one of the ways to manage pest activities in a plantation. Nevertheless, the use of synthetic insecticides causes environmental and health damage, which encourages the search for potential biopesticides [101]. The aqueous and oil extracts of soursop seeds have high larvicidal and insecticidal capacity against the vectors *Aedes albopictus* and *Culex quinquefasciatus*, with a high lethal effect in the third instar larvae and adult mosquito stages, showing greater efficiency compared to the reference insecticide deltamethrin. The extracts contain flavones, flavonones, triterpenoids, unsaturated sterols, polyphenols, and alkaloids, the latter giving insecticidal properties to the extracts [76].

Alkaloids are plant chemical compounds with industrial applications, with isoquinoline, aporphine, and protoberberine as the main ones in soursop. The larvicide potential of soursop was confirmed in the study conducted by Parthiban et al. [43]. Nonetheless, in seed without tegument, the 0.9% saline extract showed 100% mortality in fourth instar larvae of *Aedes aegypti*, *Anopheles stephensi*, and *Culex quinquefasciatus* at minimum concentrations of LC_50_: 0.024, 0.02, and 0.028 mg/mL of extract, respectively. It inhibited the enzymes acetylcholinesterase, α and β-carboxylesterase, and glutathione S-transferase, which are responsible for the detoxification of a substance in the physiological system of the insect causing the larvicidal effect of the lectin.

Therefore, soursop seeds can be used as a natural insecticide to control vectors in the environment, as well as an ecological product to minimize the toxic effects on human health caused by synthetic insecticides.

Another application of silver nanoparticles with aqueous leaf extract is the larvicide potential, in which Santhosh et al. [102] found the greatest effect against fourth instar larvae of mosquitoes *Anopheles stephensi*, *Culex quinquefasciatus*, and *Aedes aegypti*. An acute toxicity (LC_50_) of 25.47, 21.10, and 7.41, respectively, was observed in comparison with crude aqueous extract of the leaves, with LC_50_ of 458.2, 442.3, and 349.1 in 48 h. These results were similar to those found in the study conducted by Amarasinghe et al. [103] on the evaluation of larvicidal effect against *Aedes aegypti* and *Aedes albopictus* of silver nanoparticles with an extract of *Annona glabra* leaves, also belonging to the *Annonaceae*.

### 4.6. Adsorbers

Industrial activities generate several chemical pollutants daily, such as wastewater, which generally carry dyes, heavy metals, and other contaminants. It is essential to manage ways to reduce the environmental impact, preferably by a considered ecological system [104]. In this sense, the use of agro-industrial residues becomes an option.

Soursop peel, as well as other parts of the fruit, can have adsorbent effects on industrial dyes and heavy metals. Defatted powder from soursop seeds can be used also as an adsorbent for dyes from industrial effluents, such as crystal violet and methylene blue [105,106]. Ndamitso et al. [107] analyzed the use of soursop peel to remove lead, cadmium, and cobalt in oil-spill water, achieving a satisfactory result in which metal adsorption is dependent on pH, particle size, contact time, and adsorbent dosage, showing a maximum removal of 78.43%, 65.22%, and 88.75% for Cd, Co, and Pb, respectively.

Meili et al. [105] used soursop residues comprising the seeds, peel, and pulp fiber after drying at 50 °C as bio-adsorbents for the methylene blue dye. The authors achieved a removal of 91.6% in 100 mg L^−1^ of dye concentration and 80% removal at 150 mg L^−1^ for 0.75 g and 0.5 g, respectively. Therefore, soursop residues is considered a good adsorbent due to its activity and low cost.

When studying silver nanoparticles with soursop extracts, Velidandi et al. [108] used silver nanoparticles with aqueous extracts from the leaves in the degradation of rhodamine-B and methyl orange dyes considered toxic.

## 5. Projection of the Recovery of Residues from the Soursop Production Chain and Perspectives for Future Research

The conditions of waste generation without considering the composition of that waste can direct crushed, defective fruits, seeds, peels and leaves for composting and burning, in addition to disposal without any criteria. These are common destinations in producing countries and may represent more than 30% of losses in the fruit value [6].

The main soursop producing countries sell it fresh and also in the form of pulp. Data on fresh fruit prices were compared between Brazilian [109] and Mexican [110] markets, where the price of fresh fruit may vary from 1 to 10 USD/kg. The price of frozen pulp may also vary between 5 and 10 USD/kg in Brazil and Mexico, respectively [110]. Therefore, it is possible to observe that both fresh and frozen fruits have good value in these markets. With the initiative to apply waste based on the work conducted to value all parts of the fruit [59,78], the simplest processes for using residues from the soursop production chain could add opportunities to this sector, without considering those that would require more investment. Figure 2 illustrates, in a simplified way, the products that could be generated from each ton of raw material.

The proposals presented in Figure 2 combine the simplest processes based on information from the literature as well as from the world market of soursop derivatives. Under the concepts of clean technologies, each process, no matter how simple, needs to review the conditions of the waste generation site and the feasibility of on-site processing, or to establish the minimum distance between generation and processing, thus minimizing the consumption of water, energy, and the use of enzymatic extraction technologies. In addition, the replacement of solvent extractions and the feasibility of employment in the food sector should also be analyzed, according to the sanitary conditions in which the waste is obtained, allocating these materials to other sectors where the requirements are not so important, but where these can still be transformed into value-added products. Therefore, in addition to the appreciation of all parts of the fruit, other applications can be implemented for food, pharmaceutical, and cosmetic sectors, aiming to value this raw material with processes suitable for the producing regions, using new processes and minimizing the use of solvents and being less aggressive to the environment.

From one ton of soursop, fresh fruit (500–700 kg) or frozen pulp (400–600 kg) is sold. Several products can be produced, with the soursop powder being the simplest, originating about 75 kg to 125 kg of powder, with approximately 10% moisture. There is a great demand for soursop powder in the natural product market; this alternative would transform this residues into materials with different applications with high added value (Appendix A). As for the use of seeds and oil extraction cake, although the seed constituents are found in smaller percentages in the residues compared to falling and crushed fruits, the exemplified products can be elaborated by the producing communities. Likewise, the peels converted into powder could contribute to zero waste generation in this production chain. With respect to the leaves, they already correspond to a consolidated application in the production of teas, but issues of transportation and care with the alterations before the drying process must be considered. Nevertheless, it is still an important alternative from traditional applications to obtain and value the extracts.

Therefore, in the world market, soursop leaves and soursop powder are the main products, and there is a great opportunity to offer new products from soursop peels and seeds. More in-depth studies and projects on the destination of waste to more elaborate applications, such as the production of fermented alcoholic beverages, vinegar, kombucha, and biodiesel are necessary. Furthermore, a collaboration among producers is crucial, so the volumes collected, preserving the composition of the waste, justify the applications suggested in this revision.

Waste recovery must follow the same chemical and microbiological safety standards as food processing. Therefore, it is important to warn that the leaves and the peels can carry pesticides, polycyclic aromatic hydrocarbons, and other pollutants, which can be extracted by the same processes as bioactive compounds. Additionally, the type of transportation and handling of other by-products must follow all the criteria for the success of the circular economy implemented in the soursop production chain.

## 6. Conclusions

In this review, it was possible to gather, in a single work, the possibilities of valorization of all the residues of the soursop production chain. There are studies highlighting the properties of soursop by-products for human health, with great pharmaceutical and medicinal application, such as the anticancer efficacy of fruits. However, with respect to other application areas, much remains to be explored on the use of residual parts from soursop processing in the development of new products and applications, especially the studies corresponding to soursop peel.

The international market for leaves and soursop powder is consolidated; however, derivatives of seeds and peel still deserve attention. Information on the market availability of products from these residues does not indicate their effective valorization, despite the studies presenting this perspective. Fruits discarded due to crushing, excessive ripeness, unripe nature, and plant diseases, as well as seeds, peels, and soursop leaves, are still presented as residues. From natural or even dehydrated leaves, it is possible to get powdered products, which are those seen in natural food stores, but other items are not found, which shows the needed direction of future work.

The literature highlights the large losses caused by phytopathologies and deficient management, which can be minimized with more specific care in the orchard and with the harvested, selected fruit, as well as throughout the transportation to the consumer or processing units.

From this review and the presentation of the different possibilities of applications for fruit not consumed in the preparation of products, or even for raw materials outside conventional employment conditions, new opportunities can be glimpsed regarding the producing communities for the generation of new products, in addition to contributing to the reduction of waste.

## Figures and Tables

**Figure 1 foods-12-01448-f001:**
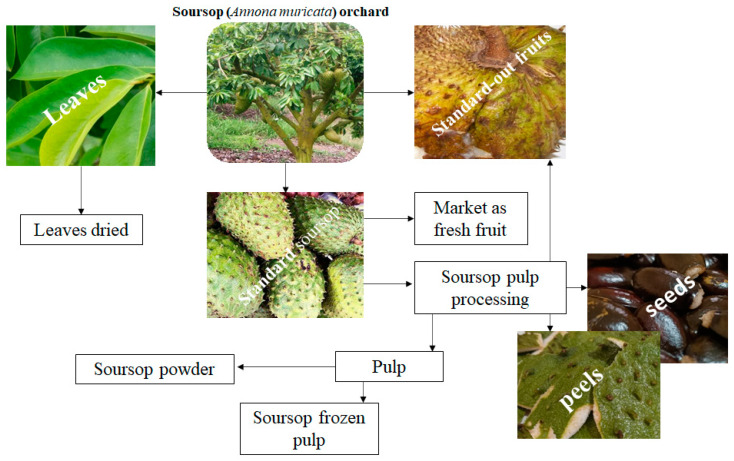
General scenario of waste generation in the soursop production chain.

**Figure 2 foods-12-01448-f002:**
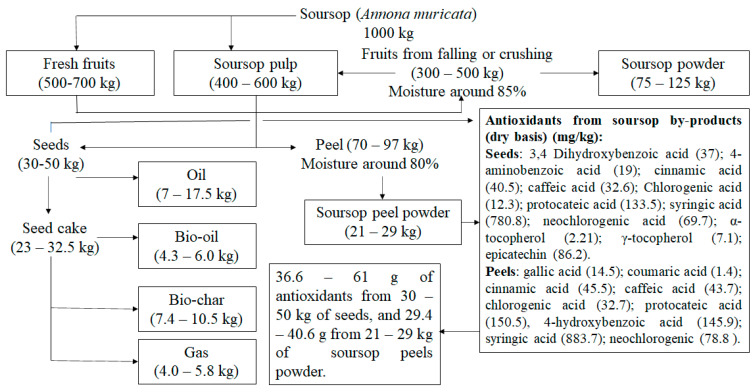
Projection for the simplest applications for the residues of the soursop production chain.

**Table 1 foods-12-01448-t001:** Bioactive compounds in soursop leaf peel, pulp, and seed.

Bioactive Compounds	Leaf	Peel	Pulp	Seed
	References			
4-Aminobenzoic acid				[20]
β-amyrin				[20]
*p*-Anisic acid				[20]
Annoionoside	[21]			
Anonaine	[21]			
Apeginin				[20]
Argentinine	[21]			
Benzoic acid		[19]		
3,4 Dihydroxybenzoic acid				[20]
4-Hydroxybenzoic acid		[19]	[19]	[20]
Blumenol C glucoside	[21]			
Caffeic acid	[22]			[19,20]
Caffeic acid derivative			[17]	
5-Caffeoylquinic acid			[17]	
Dicaffeoylquinic acid			[17]	
Carnosol				[20]
Catechin	[21,22]			[20]
Catechin gallate	[22]			
Chlorogenic acid	[21,22,23]	[19]	[19]	[19,20]
Neochlorogenic acid		[19]	[19]	[19]
Chrysin				[20]
Cinnamic acid		[19,24]	[17,19]	[19,20]
Cinnamic acid derivative	[19,20]		[17]	
Citroside A	[21]			
Coclaurine	[21]			
Coniferaldehyde				[20]
Corossolone	[21]			
Coumaric acid		[19]	[19]	[19]
Coumaric acid hexose			[17]	
*p*-Coumaric acid	[22]		[17]	[19,20]
*p*-Coumaric acid methyl ester			[17]	
Cyanidin		[24]		
9,19-cyclolanostan-3-ol,24-methlene-,(3β)				[25]
9,19-cyclolanost-24-en-3-ol,(3β)				[25]
2,8-dimethyl-2-(3E,7E)-,8,12-trimethyltrideca-3,7,11-trien-1-yl)chroman-6-ol				[25]
Datiscetin	[21]			
Ellagic acid				[20]
Epicatechin	[22,23]			[20]
Epicatechin gallate	[22]			
Epigallocatechin	[22]			
Eriodictyol				[20]
Ferulic acid	[22]			[20]
4-Feruloyl-5-caffeoylquinic acid			[17]	
Feruloyl-glucoside			[17]	
Fustin				[20]
Galangin				[20]
Gallic acid		[19]	[19]	[19,20]
Gallocatechin gallate	[22]			
Hispidulin				[20]
Kaempferol	[21]			[20]
Dihydrokaempferol-hexoside			[17]	
Kaempferol-rhamnoside	[26]			
Kaempferol 3-o-rutinoside	[21]			
Lanost-7-en-3-one,(9β, 13α, 14β,17α)				[25]
Loliolide	[21]			
Isolaureline	[21]			
Lupeol				[25]
Luteolin		[24]		
Mandelic acid				[20]
Myricetin				[20]
Naringenin				[20]
Naringin	[22]			
Norcorydine	[21]			
Pinocembrin				[20]
Procyanidin B2	[23]			
Procyanidin C1	[23]			
Protocatechuic acid		[19]	[19]	[19]
Quercetin	[21,23]	[24]		[19]
Isoquercetin	[21]			
Quercetin-diglucoside	[23]			
Quercetin-glucosyl-pentoside	[23]			
Quercetin-glucoside	[23]			
Quercetin-rhamnoside	[23]			
Quercetin-xylosyl-rutinoside	[23]			
Resorcinol		[24]		
Reticuline	[21]			
Rosmarinic acid				[20]
Rutin	[21,22,23]			[20]
Salicylic acid				[20]
Scopoletin				[20]
Sinapaldehyde				[20]
Sinapic acid				[20]
Stepharine	[21]			
Syringaldehyde				[20]
Syringic acid		[19]	[19]	[19,20]
Taxifolin				[20]
Tirucallol				[25]
Umbelliferone				[20]
Vanillic acid				[20]
Vanillin		[24]		[20]
Vitexina				[20]
Vomifoliol	[21]			
Xylopine	[21]			

**Table 2 foods-12-01448-t002:** Details of in vitro and in vivo assays of the properties of different parts of soursop (*Annona muricata*) *.

In Vitro		In Vivo	
Soursop fruit			
*Antibacterial* Methanol extract from dehydrated whole soursop fruit, *Escherichia coli* ATCC 25922, *Pseudomonas aeruginosa* ATCC 15422, *Micrococcus luteus* ATCC 4698, and *Staphylococcus aureus* ATCC 2592 [26]		*Trombolytic* Swiss albino mice of either sex (male and female), treated with crude methanol extract from dehydrated whole soursop fruit [26]	
*Antioxidant* Methanol extract of dried fruit and leaves as well as isolated 15-acetyl guanacone were evaluated for antioxidant activity by DPPH, ABTS, and ferric reducing in comparison to control (ascorbic acid) [31]		*Antidiarrheal activity* Castor oil induced method; the control group received vehicle (normal saline solution, post orally), the positive control group received loperamide, and the test group received soursop pulp methanol extracts [26]	
Soursop pulp			
*Antioxidant* DPPH radical-scavenging activity of methanol pulp extract [30]		*Antidiabetic* Methanol extracts of pulverized soursop pulp and leaf to male albino Wistar rats in different doses. At the end of the 28-day experimental period, serum was collected separately and used for serum amylase assay [27]	
*Antitumor* Human tumor cell lines from MCF-7 (breast carcinoma without over-expression of the HER2/c-erb-2 gene), SKBr3 (breast carcinoma, in which the HER2/c-erb-2 gene is overexpressed), PC3 (prostate carcinoma), and HeLa (cervix epithelial carcinoma). Human dermis fibroblasts were used as control cells. Aqueous and ethanol extracts of soursop pulp [30]		*Antifungal* Methanol and aqueous extracts (dried and applied as aqueous solution) from soursop pulp controlling blackspots of *Alternaria alternata* in tomatoes [29]	
*Antimicrobial* Agar disc diffusion method for screening the antimicrobial activity of ethanol and aqueous pulp extract against *Salmonella enterica* ser. Enteritidis, *Staphylococcus aureus*, and *Listeria monocytogenes* [30]			
*Antidiabetic* Pulp extract amylase inhibition assay; pancreatic alpha-amylase of porcine origin [27,28]			
*Antihypertensive* Angiotensin-I converting enzyme (ACE). Inhibition assay [28]			
*Antifungal* Inhibitory activity of methanol and aqueous soursop pulp extract on the radial growth of *A. alternata* [29]			
Soursop leaf			
*Antidiabetic* Pulp and leaf extract amylase inhibition assay; pancreatic α-amylase of porcine origin [27] Ethanol extract of soursop leaf evaluated about inhibitory against α-amylase, α-glucosidase, and lipase [23,27]		*Gastroprotective* Sprague Dawley strain rats (gastric injury induced) were treated with ethyl acetate extract of *A. muricata* leaves. Results evaluated by histopathology and immunohistochemistry [32]	
*Antioxidant* Methanol extract of dried leaves and isolated 15-acetyl guanacone, evaluated for antioxidant activity by DPPH, ABTS, and ferric reducing in comparison to control (ascorbic acid) [31]		*Anaesthetic* Wister albino rats and mongrel dogs were used for the study. They were induced for local and general anesthesia with different doses of soursop leaf methanol extracts [33]	
Soursop leaf nanoparticles as antioxidant assayed by DPPH, ABTS, and inhibition of lipid peroxidation [34] Soursop leaf extracts (80% methanol) and aqueous extracts were evaluated for antioxidant activity by FRAP, ABTS, DPPH, and nitrite [22] Soursop leaf ethanol extract, ORAC, FRAP, DPPH. Inhibition tests for the formation of advanced glycation end products; inhibition of non-enzymatic lipid peroxidation [23] Soursop leaf ethanol:acetic acid extract, DPPH, ABTS assay [34]			
*Antibacterial* Soursop leaf nanoparticles evaluated against *Staphylococcus aureus*, *Escherichia coli*, *Serratia marcescens*, *Bacillus cereus*, *Pseudomonas aeruginosa*, and *Salmonella enterica* by using a microdilution assay. The growth of bacterial isolates was considered as optical density at 530 nm [35]			
*Anti-angiogenic* 3-(4,5-dimethylthiazol-2-yl)− 2,5-diphenyltetrazolium bromide (MTT) dye reduction assay in microplates. The assay is dependent on the reduction of MTT by mitochondrial dehydrogenases of viable cell to a blue formazan product, measured spectrophotometrically (550 nm). Incubated with serial dilutions of aqueous or DMSO of leaf soursop extracts [21]			
*Antiparasitic* *T. gondii* proliferation, NIH/3T3 fibroblasts were cultured in well plates and infected with tachyzoites of *T. gondii* RH-2F1 strain cells and treated with different concentrations of soursop leaf extracts. Chlorophenol red–β-D-galactopyranoside was utilized for measuring the *T. gondii* growth [35]			
*Enzymes inhibition* Ethanolic extract of soursop leaves, α-amylase, α-glucosidase, and pancreatic lipase inhibition [23] *Protection and treatment of radiation-induced skin damage* Soursop leaves polysaccharides were tested as a protector of irradiated human cells (keratinocytes); evaluation of the effect by measuring cell viability and oxidant enzyme activity [36]			
*Anticancer* The various soursop leaf methanolic extracts were used to several fractions: hexane, hexane-ethyl acetate, ethyl acetate, ethyl acetate-methanol, methanol, and methanol-water. Each fraction was dissolved in dimethyl sulfoxide (DMSO) to generate the desired stock solution. Effects of these fractions on cancer cell viability [31] Ethanol extract of soursop leaf effect on liver cancer HepG2 and colon cancer HCT116 cells. Cell viability and apoptosis assays, bioinformatics, and proteomics [37]			
*Anthelmintic* Eggs to perform the egg hatch test (EHT) and for culture of infective larvae for larval motility test (LMT) were obtained from fecal samples collected rectally from a monospecifically *H. contortus* infected sheep. The effect of crude *A. muricata* leaf aqueous extract was evaluated [38]			
Soursop peel			
*Antihypertensive* Angiotensin-I converting enzyme (ACE) inhibition assay [28]		*Restoration of pancreatic cells* Aqueous extract of *Annona muricata* peels were tested in alloxan-induced diabetic male Wistar rats. Effect evaluated by biochemical parameters in serum and liver, antioxidant biomarkers, activity of glycolytic enzymes, and metabolomic analysis [24]	
*Bactericidal* The bactericidal effect of the soursop peel aqueous and ethanol extracts was evaluated with the modified Kirby–Bauer disk diffusion method. The effect on *S. aureus* ATCC25923, *Vibrio cholerae* classic 569B, *S. Enteritidis*, and *E. coli* was evaluated [39]			
Soursop seed			
*Antioxidant* DPPH radical-scavenging activity of methanol seed extract [30] Aqueous extract of soursop seed, DPPH, ABTS, and hydroxyl (OH) radical scavenging assay [28]		*Antifungal* Methanol and aqueous extracts (dried and applied as aqueous solution) from soursop seed controlling blackspots of *Alternaria alternata* in tomatoes [29]	
*Antitumor* Human tumor cell lines from MCF-7 (breast carcinoma, without over-expression of the HER2/c-erb-2 gene), SKBr3 (breast carcinoma, in which the HER2/c-erb-2 gene is overexpressed), PC3 (prostate carcinoma), and HeLa (cervix epithelial carcinoma). Human dermis fibroblasts were used as control cells. Aqueous and ethanol extracts of soursop seed [30]		*Attenuation in benign prostatic hyperplasia* Adult male Wistar rats treated with testosterone and soursop seed n-hexane extract. Effect evaluated by immunohistochemical on the expression of proteins and histology of prostate, markers of inflammation, and antioxidants [25]	
*Antimicrobial* Agar disk diffusion method for screening the antimicrobial activity of ethanol and aqueous seed extract against *Salmonella enterica* ser. Enteritidis, *Staphylococcus aureus*, and *Listeria monocytogenes* [30]		*Antiatherogenic* Male albino Wistar rats streptozotocin-induced diabetics, treated with soursop seed aqueous extract. The effect on biochemical markers was determined [40]	
*Antidiabetic* Soursop seed extract amylase inhibition assay; pancreatic alpha-amylase of porcine origin [28] *Antihypertensive* Angiotensin-I converting enzyme (ACE) inhibition assay [28]		*Antidiarrhea* Wistar albino rats with castor oil induced diarrhea were treated with soursop ethanol seed extracts. Gastro-intestinal mobility was evaluated [41]	
*Larvicidal* Acetogenin-rich fraction of *A. muricata* seeds and annonacin effects on the larvae of *Ae. aegypti* and *Ae. Albopictus* were verified from the analysis of the main enzymes of the *Culicidae* larvae metabolism [42] Soursop seed kernel powder extracts (hexane, chloroform, ethyl acetate, and ethanol). Early fourth instars of *A. aegypti*, *A. stephensi*, and *C. quinquefasciatus* larvae were introduced into water containing each solvent extract. Assay of biochemical constituents; threshold time for lethal effect was evaluated [43]			

* All cited references are from each published manuscripts cited in this work.

## Data Availability

All data are included in this manuscript.

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
