# Peer review of "Soursop (Annona muricata) Properties and Perspectives for Integral Valorization"

_foods, 2023, doi:10.3390/foods12071448_

Round 1

Reviewer 1 Report

The manuscript entitled “Soursop (Annona muricata) properties and perspectives for integral valorization”, authored by Santos and colleagues, deals with the summarizing of the main information relative to Sourpop fruits. The authors focused their attention not only on botanical information, but also on nutritional values and functional properties. The strength of this manuscript is related to the valorization of waste products, highlighting how this plant material is important not only from a nutraceutical point of view, but also from a circular economy point of view.

Overall, I am positively impressed with this manuscript. It is evident that the authors are experts in the subject, and I can only congratulate them on the structure and topics covered. The manuscript is written with authority and contains information that is truly useful to the current state of the art.  However, I would like to suggest a number of changes.

1 - The affiliations section should include each author's email, along with their acronym. This acronym should be the same one used in the contributions section.

2 - The abstract contains general information related to Soursop. However, the purpose of the Review is only hinted at in the last two lines. The authors should explain in this section the motivation for writing this manuscript, better specifying the purpose and future prospects.

3 - I fully agree with the information reported in the introduction section. Sourpop, and generally other fruits belonging to the Annonaceae family, are fruits typical of sub-tropical regions. However, one of the few benefits of the climate change that is currently affecting our planet is the development of climates favorable for growing Annonaceae plants even in regions other than South America. For example, a recent work has reported (doi.org/10.3390/foods10010035) how in Sicily and Calabria (Italy) or in Spain a climate other than the Mediterranean has developed, very similar to the sub-tropical climate. This has allowed the experimental cultivation of these plants in European territory as well. The authors should therefore specify that cultivation of these plants is also beginning in Europe. The motivation for the cultivation of Annonaceae also in European territory is because there is increasing recognition of the nutritional value, nutraceutical traits, and functional properties of these fruits that were previously completely unknown in Europe. These works (references below) have the extreme utility of enhancing the value of these fruits in foreign territory as well, and making modern consumers aware of exotic fruits they are not accustomed to.

4 - Section 3, 4, 5, and 6 should be divided into several subsections, each for each type of food-product (section 3) or application (section 4).

5 - Section 2 mentions the main potential functionalities demonstrated by Souropop and its by-products, attaching a table. It would be appropriate to better argue the biological activities, highlighting the main differences described in the different works. For example, are the phytochemical characterizations concordant in the different papers? And what about the biological activities?

6 - Some keywords should be added. The utility of these terms is to facilitate the search of the article using common scientific search engines (PubMed, GoogleScholar, Scopus, etc.), which rely on the terms contained in title, abstract, and keywords. I strongly suggest adding additional keywords and if the authors see fit replace some terms already repeated several times in the abstract.

7 - Some minor typos are present in the text, which needs correction. However, I do not think that this is a major problem that can compromise the positive judgment to this manuscript.

8 - Authors should also pay attention in the style and format used. it is true that styling problems are corrected and fixed by the editorial team during the late stages of submission, and only in case the manuscript is accepted, however, a bad formatting can negatively affect the Reviewers' opinion (this is not my case).

9 - Figure captions found on page 22 are not necessary.

10 - Reference style and formatting should be changed according to the guidelines of the journal.

11 - Page 23 is redundant.

Author Response

Dear reviewer,

We are very grateful for all the suggestions that may contribute to the work reaching the quality standard required by Foods.

Corrections are highlighted in the text in red letters.

1-The affiliations section should include each author's email, along with their acronym. This acronym should be the same one used in the contributions section.

Author’s email were included.

2 – Abstract

The authors should explain in this section the motivation for writing this manuscript, better specifying the purpose and future prospects.

Was corrected.

3 -  Introduction

 “…one of the few benefits of the climate change that is currently affecting our planet is the development of climates favorable for growing Annonaceae plants even in regions other than South America.”

The statement was included in the introduction, with the reference cited.

4 - Section 3, 4, 5, and 6 should be divided into several subsections, each for each type of food-product (section 3) or application (section 4).

The sections were modified, trying to meet the request, which made the presentation more fluid.

5 - Section 2 mentions the main potential functionalities demonstrated by Souropop and its by-products, attaching a table. It would be appropriate to better argue the biological activities, highlighting the main differences described in the different works. For example, are the phytochemical characterizations concordant in the different papers? And what about the biological activities?

Key information on phytochemical compounds and bioactive properties has been highlighted in Table 1 and in the text.

Table 1 has been modified and a new table has been added.

6 - Some keywords should be added. The utility of these terms is to facilitate the search of the article using common scientific search engines (PubMed, GoogleScholar, Scopus, etc.), which rely on the terms contained in title, abstract, and keywords. I strongly suggest adding additional keywords and if the authors see fit replace some terms already repeated several times in the abstract.

Keywords added: Annona muricata; Bioinsecticides; Biorefinery; Pharmaceuticals; Circular bioeconomy; and  Phytochemicals.

7 - Some minor typos are present in the text, which needs correction. However, I do not think that this is a major problem that can compromise the positive judgment to this manuscript.

The text has been subjected to a new English proofreading.

8 - Authors should also pay attention in the style and format used. it is true that styling problems are corrected and fixed by the editorial team during the late stages of submission, and only in case the manuscript is accepted, however, a bad formatting can negatively affect the Reviewers' opinion (this is not my case).

We paid more attention to formatting and styling as suggested.

9 - Figure captions found on page 22 are not necessary.

Was removed.

10 - Reference style and formatting should be changed according to the guidelines of the journal.

Was corrected

11 - Page 23 is redundant.

Was removed.

On behalf of all authors, I thank you once again for your consideration of our work and forward the manuscript with the suggestions incorporated, including the suggestions and corrections of the two reviewers.

Best regards,

Edna Regina Amante.

Reviewer 2 Report

The current review entitled "Soursop (Annona muricata) properties and perspectives for integral valorization" gives some information on properties of soursop (Annona muricata) by-products which is suitable to the sustainability of productive chain by-products. Eventhough the topic is interesting but it needs major revisions before the publication. As an example and from my perspective, all the Figures should be structurally changed to be more engaging  and give the readers more information. My specific comments are shown in the PDF file.

Thank you 

Author Response

Response to Reviewer #2

Dear reviewer,

On behalf of the authors, we thank you for your consideration and attention to our work.

We started to answer the questions and sent a copy of the manuscript containing your considerations highlighted in red letters..

Abstracts

“Tropical countries are examples....”

Was corrected.

Introduction

According to Alves and Lima, precise statistical data regarding production numbers are difficult to obtain..., and this was included in the work.

The sentence has been rewritten

“Fresh fruit is the market's favorite, as it has a pleasant and sweet taste, but the sensitive characteristics of the fruit lead to the production of pulp as the most economically attractive commercial form.”

The term circular economy was introduced.

“…objective offer information that can contribute to future biorefinery projects and to the circular economy in the soursop production chain.”

On the influence of different soursop varieties:

“Just as the different varieties interfere with the properties of the pulp, it is important to consider these differences in terms of residues in the soursop production chain.”

Table 2 was introduced in order to detail the in vivo and in vitro studies in the different works.

Linolieic was removed.

Figures 2 to 5 were removed and a new Figure 1 was constructed.

Thanks again for all the considerations.

Best regards,

Edna Regina Amante.

Round 2

Reviewer 2 Report

The authors have provided enough responses to my comments

and the current version of the manuscript can be considered for the publication.

Thank you